

# Merging with crowdsourced rain gauge data improves pan-European radar precipitation estimates

Aart Overeem[1], Hidde Leijnse[1], Gerard van der Schrier[1], Else van den Besselaar[1], Irene Garcia-Marti[1], and Lotte Wilhelmina de Vos[2]

[1]R&D Observations and Data Technology, Royal Netherlands Meteorological Institute, Utrechtseweg 297, 3731 GA De Bilt, The Netherlands
[2]Observation Operations, Royal Netherlands Meteorological Institute, Utrechtseweg 297, 3731 GA De Bilt, The Netherlands

**Correspondence:** Aart Overeem (aart.overeem@knmi.nl)

**Abstract.** Ground-based radar precipitation products typically need adjustment with rain gauge accumulations to achieve a reasonable accuracy. This is certainly the case for the pan-European radar precipitation products. The density of (near) real-time rain gauge accumulations from official networks is often relatively low. Crowdsourced rain gauge networks have a much higher density than conventional ones and are a potentially interesting (complementary) source to merge with radar

precipitation accumulations. Here, a 1-year personal weather station (PWS) rain gauge dataset of ∼5 min accumulations is obtained from the private company Netatmo over the period September 1, 2019–August 31, 2020, which is subjected to quality control using neighbouring PWSs and on 1-h accumulations using unadjusted radar data. The PWS 1-h gauge accumulations are employed to spatially adjust OPERA radar accumulations, covering 78% of geographical Europe. The performance of the merged dataset is evaluated against daily and disaggregated 1-h gauge accumulations from weather stations in the European

Climate Assessment & Dataset (ECA&D). Results are contrasted to those from an unadjusted OPERA-based radar dataset and from EURADCLIM. The severe average underestimation for daily precipitation of ∼28% from the unadjusted radar dataset diminishes to ∼3% for the merged radar-PWS dataset. A station-based spatial verification shows that the relative bias in 1-h precipitation is still quite variable and suggests stronger underestimations for colder climates. A dedicated evaluation with scatter density plots reveals that the performance is indeed less good for lower temperatures, which points to limitations in

observing solid precipitation by PWS gauges. The outcome of this study confirms the potential of crowdsourcing to improve radar precipitation products in (near) real-time.

## 1   Introduction

Accurate precipitation information is required for many applications and scientific disciplines, such as weather monitoring and nowcasting, climate monitoring, water management, (flash) flood modelling and forecasting, extreme value modelling and

evaluation of extreme precipitation events, hydropower generation and agricultural production. Official rain gauge networks often do not capture the spatial variability in precipitation because of too low network densities, especially in (near) real-time and in urban areas. Moreover, data from these networks are not always openly available. Ground-based weather radars provide wide coverage and (near) real-time information at the kilometre scale every 5 or 10 minutes, but their precipitation estimates can





be influenced by many sources of error (Joss and Waldvogel, 1990; Fabry, 2015; Rauber and Nesbitt, 2018). These can be much
improved by additional processing, especially in case of dual-polarisation radars. However, these algorithms have not always
been tested, let alone applied in operational products. Moreover, (large) deviations in quantitative precipitation estimation
(QPE) with respect to true precipitation received at the Earth's surface can still occur. Hence, it is common practice to combine
rain gauge and radar data for QPE (e.g. Goudenhoofdt and Delobbe, 2016; Nelson et al., 2016; Winterrath et al., 2018; Yu
et al., 2020), also called gauge-adjusted radar QPE. Often rain gauge and radar data from the same interval are combined,
called merging. In case gauge data are lacking, an adjustment factor field can be computed based on (recent) historical radar
and gauge data and subsequently applied to current radar data (Park et al., 2019). Increasing the density of rain gauge networks
will generally improve merged radar precipitation products, compensating for the spatially and temporally varying sources of
errors in radar QPE. This does require rain gauge data of sufficient quality.

A novel source of in-situ surface rainfall measurements are crowdsourced or third-party personal weather stations (PWSs)
with rain gauges. The high densities of these kind of Internet of Things (IoT) rain gauge networks and their potential real-
time availability, hold a promise for improving merged (real-time) radar-gauge products, especially at the sub-daily time scale.
PWSs are not expected to yield the same observational quality compared to official automatic weather stations. This is not only
influenced by the quality of PWS sensors, but setup, surrounding environment, metadata, connectivity, data transfer, power
supply, calibration, and maintenance also play crucial roles in this (De Vos et al., 2017; Bárdossy et al., 2021; Hahn et al.,
2022). De Vos et al. (2017) obtain highly accurate rainfall estimates (compared with collocated official rain gauge data) in
an experimental setup with three properly installed Netatmo PWS rain gauges, suggesting that sensor quality can be good.
The hypothesis is that after thorough quality control (QC), the high density of a PWS network can compensate for its lower
accuracy in case interpolated or merged datasets are derived. PWSs can capture local precipitation variations, which would
otherwise go unnoticed by official gauge networks. QC algorithms have been developed for and (successfully) tested on PWS
rain gauge data (De Vos et al., 2019; Båserud et al., 2020; Bárdossy et al., 2021). Some of these have not necessarily been
designed for use on crowdsourced data (Båserud et al., 2020; Bárdossy et al., 2021) or have even been specifically developed
for conventional networks (Ośródka et al., 2022). Often, these algorithms apply similar processing steps as in De Vos et al.
(2019), i.e., removing faulty zeroes, high influx and station outliers employing inter-station checks. For instance, the open-
source R package Titan and its C++ version Titanlib consist of 12 automatic QC checks, emphasising spatial, mainly inter-
station, checks. It can use auxiliary data from, e.g., radars or numerical weather prediction models, for a first-guess check
(Båserud et al., 2020; Lussana et al., 2020; Nipen et al., 2022). It has been developed for QC of temperature and rain, although
it has mainly been tested on temperature. Titan is being employed operationally for QC of PWS air temperature data, which
are used for postprocessing of short-term weather forecasts (Nipen et al., 2020). Aiming to improve interpolated gauge-based
precipitation maps, Bárdossy et al. (2021) present another inter-station QC algorithm, which discards Netatmo PWS rain
gauge data when their distributions do not comply with those from rain gauge observations from the German Weather Service,
the primary network. This approach may not work as good for primary networks with lower densities and its (near) real-
time application would depend on the availability of primary network data. Ośródka et al. (2022) present the RainGaugeQC
algorithm, optimised for 10-min accumulations and containing five checks, consisting of inter-station and some intra-station





checks and partly involving radar data. The gross error check and radar conformity check from Ośródka et al. (2022) show
similarities with the QC in this study.

The added value of Netatmo PWS gauges for hourly to multi-day rainfall mapping compared to official rain gauges has been demonstrated for Germany, and merging of PWS gauge with radar precipitation data is considered an opportunity which could be beneficial for hydrometeorological applications (Graf et al., 2021). To the best of our knowledge, the merging of radar and PWS gauge data has not been addressed yet in the peer-reviewed literature. Here, we investigate the potential of PWS rain gauge data for one of its most important use cases: improving radar QPE. This is performed at an unprecedented scale, covering 78% of geographical Europe over a full year. A pan-European radar dataset, based on a product from the Operational Program on the Exchange of weather RAdar information (OPERA), is employed as starting point (Huuskonen et al., 2014; OPERA, 2022). This unadjusted dataset underestimates precipitation by, on average, 28% over the period September 1, 2019–August 31, 2020. The focus of this paper is not on the quality of PWS data in relation to those from the National Meteorological and Hydrological Services (NMHSs) or on quality control procedures for PWS data as such, but on the performance of a merged radar-PWS dataset. Part of the QC by De Vos et al. (2019) and De Vos (2021) is applied and additional QC, also involving unadjusted radar data, is developed. A spatial adjustment is applied to merge European-wide OPERA 1-h radar precipitation accumulations at a 2-km grid with 1-h PWS rain gauge accumulations obtained from Netatmo. The merged dataset is evaluated against rain gauge accumulations from the European Climate Assessment & Dataset (ECA&D). Results are also contrasted to those from the European climatological high-resolution gauge-adjusted radar precipitation dataset EURADCLIM (EUropean RADar CLIMatology) (Overeem et al., 2023). In order to assess the performance in case solid precipitation is likely, the evaluation is also performed separately for air temperatures below and above 5°C.

In Section 2, the employed radar precipitation datasets, ECA&D rain gauge dataset, PWS rain gauge dataset, and gridded mean daily temperature dataset are described. In Section 3, the QC algorithm of PWS data and the algorithm to merge radar and PWS data are described. In Section 4, evaluations of unadjusted OPERA and gauge-adjusted OPERA datasets against ECA&D rain gauge accumulations are provided. The performance is evaluated for lower and higher air temperatures. Annual precipitation is compared to EURADCLIM. The potential for extreme (urban) precipitation monitoring employing a merged radar-PWS dataset is illustrated. Section 5 discusses limitations and provides recommendations. Finally, Section 6 provides the main conclusions.

## 2 Data

### 2.1 Pan-European radar datasets

The radar datasets are based on the composite product "instantaneous surface rain rate", as downloaded from the EUMETNET OPERA Data Centre (ODC or Odyssey) from the period September 1, 2019–August 31, 2020. They have a grid cell size of 2 km × 2 km and are available every 15 min. The median and average number of contributing radars in the period September 1, 2019–August 31, 2020 is 154. Algorithms to take out non-meteorological echoes, including Doppler clutter filtering, and to correct for beam-blockage have been applied by NMHSs and/or OPERA. Different kind of radars are employed with,



for instance, single-polarisation and dual-polarisation radars, and most radars operate in the C-band frequency range. Radars provide data every 5 min or 10 min, where the most recent data are employed in the OPERA 15-min composite product. OPERA converts the horizontally polarised radar reflectivity factors to instantaneous surface rain rates using the Marshall-Palmer $Z_\mathrm{h} - R$ relation ($Z_\mathrm{h} = 200R^{1.6}$). The OPERA radar data and the applied algorithms are described in more detail by Saltikoff et al. (2019) and Overeem et al. (2023).

Here, two OPERA-based clock-hour radar precipitation datasets are utilised. The processing of these datasets and its characteristics are provided by Overeem et al. (2023). For each clock-hour and grid cell, accumulations are only computed in case of full availability of the underlying 15-min data. Both datasets have undergone additional removal of non-meteorological echoes by means of two statistical methods (Gabella and Notarpietro, 2002; Wradlib, 2021, 2022) and a satellite cloud type mask. The latter sets rain rates to zero in case of semitransparent clouds or cloud-free areas. Not all of these steps can be applied in real-time, but applying them yields a dataset which has undergone the same processing as the EURADCLIM dataset (see below) that is used in comparisons. Moreover, these steps only consider the removal of non-meteorological echoes and not the precipitation retrieval itself. The first dataset is an unadjusted OPERA radar dataset, called "OPERA" in this paper and corresponding to "Gabella + CTM + static filter" in Overeem et al. (2023). The second dataset is obtained by merging with disaggregated 1-h ECA&D rain gauge accumulations employing a local mean-field bias adjustment succeeded by a spatial adjustment. For this, the original daily gauge accumulations are disaggregated to 1-h accumulations using the 1-h and 24-h OPERA radar accumulations. An important reason for the disaggregation is the fact that the measurement interval of daily rain gauge accumulations is not the same for the European NMHSs, which is reflected in ECA&D as the NMHSs are the main source of data. A daily adjustment would therefore be less appropriate (Overeem et al., 2023). This publicly accessible climatological gauge-adjusted product is called "EURADCLIM" (Overeem et al., 2022a, b). A flowchart of the radar and gauge data processing for OPERA and EURADCLIM can be found in Overeem et al. (2023). From these 1-h precipitation accumulations, 24-h (every clock-hour) and annual precipitation accumulations are derived in case a grid cell has at least 83.3% availability. The combined radar-gauge availability of daily and 1-h precipitation accumulations is usually high (Figure 1a–b).



**Figure 1.** Map of Europe with combined radar-gauge availability over the period September 1, 2019–August 31, 2020 based on daily ECA&D accumulations (a) and based on 1-h PWS accumulations (b). Map of distance to nearest rain gauge per OPERA radar grid cell assuming full availability of radar and gauge data for ECA&D gauge (c) and Netatmo gauge (d) locations. This shows the best possible result. Note that the Netatmo maps are based on quality-controlled (faulty zeroes and high influx filters) data, and the combined radar-gauge availability map is based on a static radar coverage map. In reality, the average minimum distance will be longer because sometimes gauge data are missing. For maps (a) and (b), the average availability of gauges has been computed for non-overlapping square lattices of 9 radar grid cells a side (to allow for appropriate plotting of the high-density PWS dataset). The value for each square lattice is plotted in case data from at least one rain gauge is available. Maps made with Natural Earth. Free vector and raster map data ©naturalearthdata.com. https://www.netatmo.com/weather





## 2.2 ECA&D rain gauge data

Daily precipitation time series from 6,678 rain gauges were obtained mid June 2022 from the ECA&D (https://www.ecad.eu) project (Klein Tank and coauthors, 2002; Klok and Klein Tank, 2008) for the period September 1, 2019–August 31, 2020. For most regions, the combined availability of radar and gauge data is at least 90% (Figure 1a). Figure 1c displays the distance from a radar grid cell to the nearest gauge assuming full availability of the 6,678 gauges. It reveals a large spatial variability of the gauge network density. The median and mean distances for a radar grid cell to the nearest rain gauge are 42 km and 92 km, respectively. The relatively large difference between median and mean distance is primarily caused by areas with low rain gauge network densities (e.g., above sea). Three radar precipitation datasets will be evaluated with the ECA&D rain gauge dataset. It will also be merged with 1-h radar accumulations in case of the EURADCLIM dataset. The ECA&D team (Project team ECA&D, Royal Netherlands Meteorological Institute KNMI, 2021) has applied quality control on the rain gauge data. Quality control has often also been applied by the NMHS that has provided the data. More details can be found in Overeem et al. (2023). The quality of EURADCLIM is lower for areas with low ECA&D gauge network density. Specifically for this 1-year period, some gauge data are not available for the full period. The most notable example of this is that the data from most stations in the United Kingdom end on December 31, 2019 (i.e., only 4 out of 12 months are available).

## 2.3 PWS rain gauge data

A Netatmo PWS contains an indoor and an outdoor module measuring air temperature, relative humidity, air pressure, and some additional indoor variables. A wind and rain module can be added. The rain gauge is of the tipping bucket type containing a collection funnel with a ∼13-cm cross-section. A radio connection is used to send the number of tips to the indoor module, which transfers the PWS measurements to the Netatmo platform every ∼5 min. Data are visualised at the publicly accessible Netatmo Weathermap (https://weathermap.netatmo.com/), but can also be accessed by weather station owners via smartphone or tablet. The default tipping bucket volume, and hence measurement resolution, is 0.101 mm. This can be altered by a weather station owner using a calibration feature. The manufacturer's specifications report a measurement range of 0.2–150 mm h$^{-1}$ and an accuracy of 1 mm h$^{-1}$ (Netatmo, 2022). Typical sources of error in (PWS) rain gauge observations are described in De Vos et al. (2017), De Vos et al. (2019) and Ośródka et al. (2022) and references therein.

A 1-year PWS rain gauge dataset of ∼5 min accumulations was obtained from Netatmo. Their locations are shown in Figure 1b (after applying part of the QC; see Section 3.1). The combined radar-gauge availability is generally at least 70%, averaged over square lattices of 9 radar grid cells a side. Based on the individual PWS time series, the median and mean availability is 82.8% and 69.9%, respectively. One of the reasons for this relatively low mean could be a strong increase in the number of PWSs over the course of the 1-year period (∼23%). The total number of PWSs is more than 10 times higher than that for the ECA&D rain gauge dataset. The median and mean distances for a radar grid cell to the nearest rain gauge are 28 km and 61 km, respectively. As a consequence, the distance to the nearest gauge is generally much shorter compared to the ECA&D dataset, but is still quite variable in space (Figure 1c–d). It is apparent from this figure that the network density is much higher in regions where the ECA&D gauge network density is (relatively) sparse (e.g., parts of the Iberian peninsula and eastern



Europe), or where rain gauge data are not shared with ECA&D (e.g., Bulgaria). For some regions, the Netatmo gauge network density is lower, though (e.g., in northern Scandinavia). Note that the PWS network tends to be of high density in the areas

where the population density is high, which contrasts with the networks operated by the NMHSs that generally operate their rain gauges in open rural areas (following World Meteorological Organization regulations).

## 2.4   Gridded air temperature data (E-OBS)

A pan-European gridded dataset of daily mean air temperature was obtained (E-OBS version 26.0e) to evaluate the performance of merged radar datasets for lower and higher temperatures at the location of ECA&D rain gauges. This dataset is derived from

ECA&D station observations of 2-m air temperature that are interpolated onto a 0.1° grid (Cornes et al., 2018; Copernicus Climate Change Service, 2022). Because not all stations with rain gauges have temperature sensors, the E-OBS dataset was employed to derive temperature for each rain gauge location.

## 3   Methodology

The flowchart in Figure 2 shows the employed input datasets, the applied QC to Netatmo PWS data, the merging with 1-h radar

accumulations and the output dataset "OPERA + NETATMO".

## 3.1   QC of PWS rain gauge data

The PWS measurements are preprocessed to obtain precipitation accumulations at regular 5-min intervals. This is followed by QC, where for each PWS a neighbour list of PWSs is constructed according to De Vos et al. (2019). The neighbour selection has been slightly modified: only the 20 nearest stations within a radius of 10 km are selected instead of all stations within a

10-km radius. The 20 nearest stations are expected to be more representative for QC of rainfall at the considered gauge location.

    Next, the faulty zeroes and high influx filters are computed and applied to the 5-min data. Faulty zeroes can be caused by obstruction of the tipping bucket mechanism or other malfunctioning of the gauge. Unrealistically large accumulations (high influx) can be caused by, for instance, sprinklers, tilting of the gauge or cleaning by pouring liquids into the gauge. The filters and their parameter values have been tested on a 1-year Netatmo rain gauge dataset from the Amsterdam metropolitan area,

the Netherlands, from June 1, 2016–31 May 2017 (De Vos et al., 2019). The basic principle of the faulty zeroes and high influx filter is that rainfall is correlated in space. If less than 5 neighbouring stations are available, there is insufficient basis to attribute a faulty zero or high influx flag and the value is kept (so-called flex filtering). This is done to keep rainfall values in areas with lower gauge network densities. Subsequently, only PWS clock-hour values are computed from the 5-min accumulations in case at least 10 out of 12 intervals have valid data. De Vos et al. (2019) provide a more detailed explanation of the faulty zeroes and

high influx filters, including flowcharts.





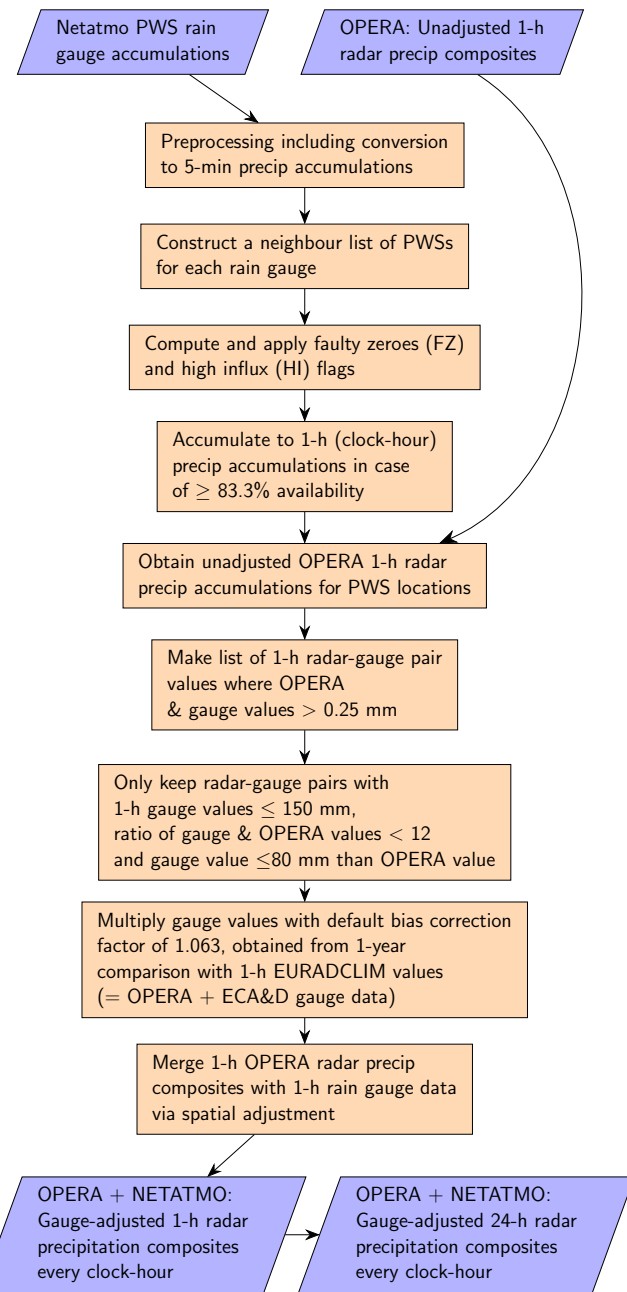

**Figure 2.** Flowchart of Netatmo PWS quality control and merging of PWS rain gauge data with OPERA radar accumulations.

Radar-gauge pairs are obtained by collocating the PWS accumulations with the unadjusted OPERA radar accumulations. Only the 1-h PWS accumulations are used for merging if both the unadjusted radar value and the Netatmo gauge value are





larger than 0.25 mm, which leads to a strong decrease in the number of radar-gauge pairs. The reason for this is twofold: it strongly reduces the computational time of the merging and the radar data also act as an additional QC. When precipitation is
weak or non-existent according to PWS or radar, the PWS values are considered erroneous. This makes application of the faulty zeroes filter largely redundant, except that it results in ∼4% fewer 1-h PWS values. Both the faulty zeroes and the high influx filters are applied to 5-min data, which may result in a more refined removal of erroneous values compared to QC on 1-h data. In addition, the 0.25-mm radar threshold can also remove outliers (high-influx) in PWS gauge values. Note that employing unadjusted radar data is justified for the thresholding, because it is meant to discern between low and high accumulations, for
which high accuracy is not needed. Finally, the effect of removing low values on the outcome of the merging is likely small.

    Only radar-gauge pairs with 1-h PWS gauge accumulations ≤150 mm are kept, which corresponds with the manufacturer's specifications (Netatmo, 2022). Next, the unadjusted OPERA radar data are employed for additional QC (radar QC). Only those 1-h radar-gauge pairs are kept if the ratio of a PWS gauge and OPERA accumulation is lower than 12 and the PWS value is at most 80 mm higher than the OPERA value. Finally, the 1-h PWS gauge accumulations are compared with the
gauge-adjusted EURADCLIM dataset, to obtain a default bias correction factor, methodologically similar to De Vos et al. (2019). This introduces some dependence, but note that only a constant default bias correction factor is obtained over entire Europe and over a full year. All 1-h PWS gauge accumulations are multiplied by this bias factor of 1.063 to compensate for underestimation.

### 3.2 Radar-PWS merging

The merging method is extensively explained by Overeem et al. (2023) for the EURADCLIM dataset. Here, the same approach is followed, except that the first step of the merging method, the local mean-field bias adjustment, is discarded. The necessity of this is much lower because of the much higher gauge network density. Moreover, it resulted in outliers. The basis of the resulting merging algorithm is the Barnes' Objective Analysis (Barnes, 1964). An adjustment factor is calculated for each radar grid cell and 1-h interval. For a given grid cell, only radar-gauge pairs within a given distance are taken into account for the
computation of the adjustment factor (only the short range component is employed Overeem et al., 2023). The value for this distance is the range of an isotropic spherical variogram model based on a 30-year rain gauge dataset from the Netherlands from the period 1979–2009 (Van de Beek et al., 2012).This range varies as a function of day of year. The adjustment factor field is applied to the OPERA 1-h accumulations to obtain the OPERA-Netatmo merged radar dataset. The median and mean number of radar-gauge pairs used in the merging, so after QC and thresholding, is ∼2,400 and ∼3,300, respectively (computed
over the ∼99.5% of 1-h intervals in the 1-year period that have at least one valid radar-gauge pair). Note that the median number of radar-gauge pairs is ∼6 times larger than for the EURADCLIM dataset, which uses ECA&D rain gauge data. The maximum number of radar-gauge pairs for a given hour is 22,866.

### 3.3 Evaluation metrics

The following statistical metrics are employed to evaluate the performance of radar precipitation accumulations: the relative
bias compared to the corresponding ECA&D gauge precipitation accumulations, the residual standard deviation divided by the





**Table 1.** Performance of radar daily precipitation accumulations over the period September 1, 2019–August 31, 2020 for all radar-gauge pairs and for those above different thresholds. ECA&D rain gauge data at their default measurement interval are used as reference. The threshold value and the mean daily precipitation are based on the gauge data. Subsequently, the relative bias, Pearson correlation coefficient, mean absolute error, coefficient of variation and number of radar–gauge pairs are provided, respectively.

| Threshold value (mm) | Mean daily precipitation | Rel. bias (%) | $\rho$ | MAE (mm) | CV | No. of pairs |
|---|---|---|---|---|---|---|
| OPERA: | | | | | | |
| | 2.56 | -28.4 | 0.78 | 1.43 | 1.55 | 2,258,828 |
| 1.0 | 7.52 | -34.7 | 0.69 | 3.80 | 0.85 | 746,753 |
| 10.0 | 19.35 | -45.3 | 0.53 | 9.85 | 0.54 | 166,721 |
| 20.0 | 32.15 | -51.7 | 0.46 | 17.59 | 0.45 | 49,783 |
| OPERA + Netatmo No QC: | | | | | | |
| | 2.56 | -4.8 | 0.84 | 1.15 | 1.32 | 2,258,825 |
| 1.0 | 7.52 | -12.1 | 0.81 | 2.82 | 0.70 | 746,753 |
| 10.0 | 19.35 | -20.3 | 0.71 | 6.25 | 0.46 | 166,721 |
| 20.0 | 32.15 | -24.8 | 0.66 | 10.53 | 0.39 | 49,783 |
| OPERA + Netatmo QC: | | | | | | |
| | 2.56 | -3.4 | 0.88 | 1.13 | 1.15 | 2,258,825 |
| 1.0 | 7.52 | -10.3 | 0.83 | 2.77 | 0.65 | 746,753 |
| 10.0 | 19.35 | -18.6 | 0.73 | 6.08 | 0.44 | 166,721 |
| 20.0 | 32.15 | -23.5 | 0.66 | 10.22 | 0.39 | 49,783 |

mean ECA&D gauge precipitation accumulation (i.e., the coefficient of variation, CV), the Pearson correlation coefficient ($\rho$) or its squared value ($\rho^2$; the coefficient of determination), and the mean absolute error (MAE). Here, a residual is the radar accumulation minus the ECA&D gauge accumulation. In some cases, leave-one-out statistics are calculated for an independent evaluation of the EURADCLIM dataset. For these leave-one-out values the adjusted 1-h radar precipitation accumulation
is calculated for a given gauge location without employing it in the adjustment. This is performed for each gauge location separately.

## 4   Results

### 4.1   Evaluation

The daily and 1-h radar precipitation accumulations are verified against ECA&D rain gauges in Tables 1 and 2, respectively, to
assess the effect of the adjustment employing PWS gauge data. When no QC is applied to Netatmo gauge data, the performance of the merged radar dataset is already clearly better than that of the unadjusted OPERA dataset. The underestimation strongly



**Table 2.** Performance of radar hourly precipitation accumulations over the period September 1, 2019–August 31, 2020 for accumulations above different thresholds. Disaggregated ECA&D rain gauge data at their default measurement interval are used as reference. The threshold value and the mean hourly precipitation are based on the gauge data. Subsequently, the relative bias, Pearson correlation coefficient, mean absolute error, coefficient of variation and number of radar–gauge pairs are provided, respectively.

| Threshold value (mm) | Mean hourly precipitation | Rel. bias (%) | $\rho$ | MAE (mm) | CV | No. of pairs |
|---|---|---|---|---|---|---|
| OPERA: | | | | | | |
| 0.25 | 1.20 | -35.7 | 0.71 | 0.61 | 0.90 | 4,361,375 |
| 5.00 | 7.99 | -44.3 | 0.56 | 4.20 | 0.51 | 105,484 |
| 10.00 | 15.06 | -42.9 | 0.48 | 7.84 | 0.47 | 17,583 |
| OPERA + Netatmo No QC: | | | | | | |
| 0.25 | 1.20 | -13.7 | 0.75 | 0.50 | 0.89 | 4,361,375 |
| 5.00 | 7.99 | -25.1 | 0.58 | 3.15 | 0.51 | 105,484 |
| 10.00 | 15.06 | -29.0 | 0.49 | 6.20 | 0.47 | 17,583 |
| OPERA + Netatmo QC: | | | | | | |
| 0.25 | 1.20 | -11.6 | 0.79 | 0.49 | 0.81 | 4,361,375 |
| 5.00 | 7.99 | -22.9 | 0.59 | 3.10 | 0.51 | 105,484 |
| 10.00 | 15.06 | -26.9 | 0.50 | 6.08 | 0.47 | 17,583 |

decreases. Values for $\rho$, MAE and CV improve, especially for larger precipitation accumulations (except for CV for 1-h accumulations). Using quality-controlled Netatmo accumulations in the merging leads to an additional improvement.

Findings are summarised for daily precipitation with QC (Table 1): 1) the strong average underestimation of ∼28% di-
minishes to ∼3%; 2) more extreme daily precipitation underestimates are greatly reduced but not eliminated; 3) correlation increases, especially for more extreme precipitation; 4) the values for CV and MAE clearly decrease for all thresholds. The scatter density plots (Figure 3a–c) align much better along the 1:1 line compared to the unadjusted dataset. Its performance is lower than for the EURADCLIM dataset, though. This is partly caused by the fact that the same gauges used for verification are part of the EURADCLIM dataset and hence this verification is not independent. The scatter density plots also reveal how
quality control has a positive impact: a group of large precipitation accumulations in case of lower gauge accumulations is removed.

For hourly rainfall (Table 2) the underestimation decreases from ∼36% for OPERA to ∼12% for the OPERA-Netatmo dataset with QC, if the ECA&D gauge accumulation is above 0.25 mm. The improvement in the value of $\rho$ is moderate. The value of CV only improves for a gauge threshold of 0.25 mm. The values for MAE show a clear decrease. The scatter density
plots for 1-h accumulations (Figure 4) show that for the merged radar datasets the values are closer to the 1:1 line. Again, the QC helps to remove the group of high merged OPERA-Netatmo values for low ECA&D gauge accumulations. Although the impact of QC on overall statistics is small, this is an important improvement. The values for CV and $\rho^2$ for the QC'ed



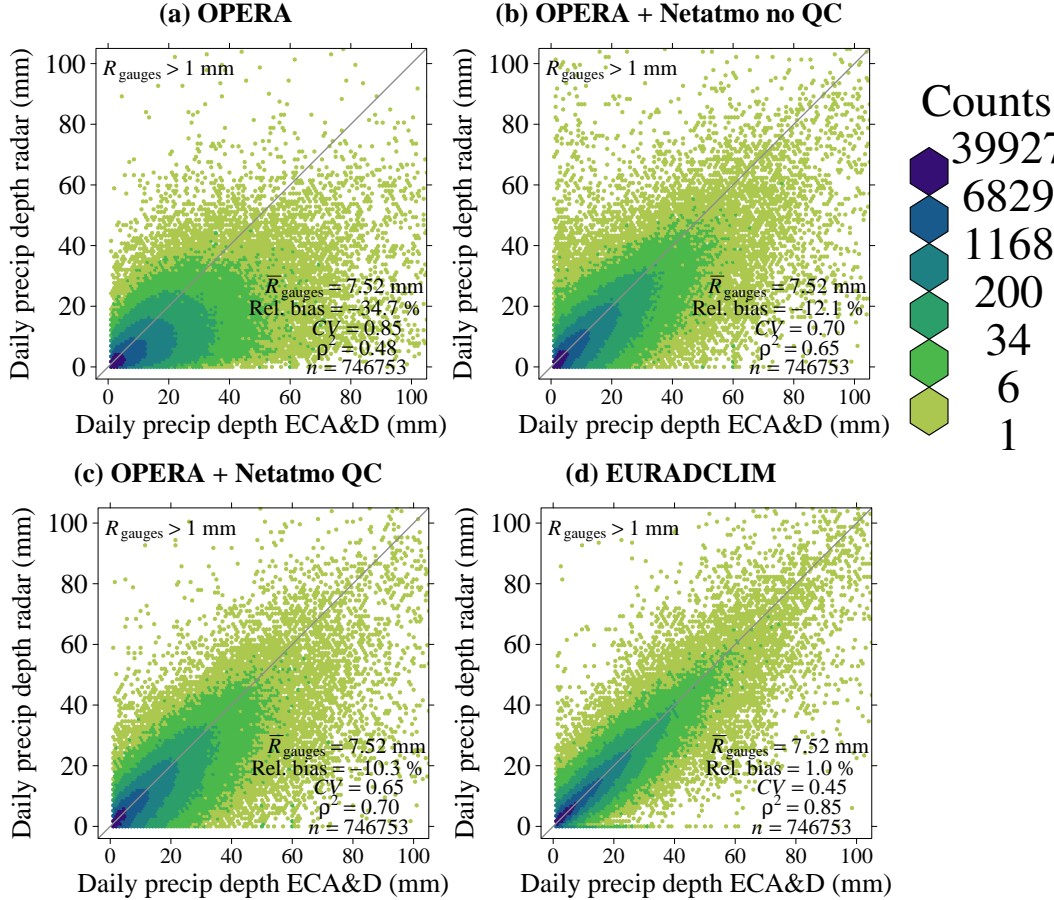

**Figure 3.** Scatter density plots of daily radar precipitation accumulations against ECA&D rain gauges (default measurement interval) over the period September 1, 2019–August 31, 2020. Results are shown for the unadjusted dataset (a), for the merged OPERA-Netatmo datasets without (b) and with (c) quality control (QC) on Netatmo gauge data, and for the gauge-adjusted EURADCLIM dataset (d). Note that the latter is not an independent verification.

OPERA-Netatmo dataset are (relatively) close to those from EURADCLIM. Note that leave-one-out statistics have been used for EURADCLIM, which is an independent verification. The only dependence for all four datasets is the use of 1-h and 24-h

unadjusted OPERA radar accumulations to disaggregate daily to hourly gauge accumulations.



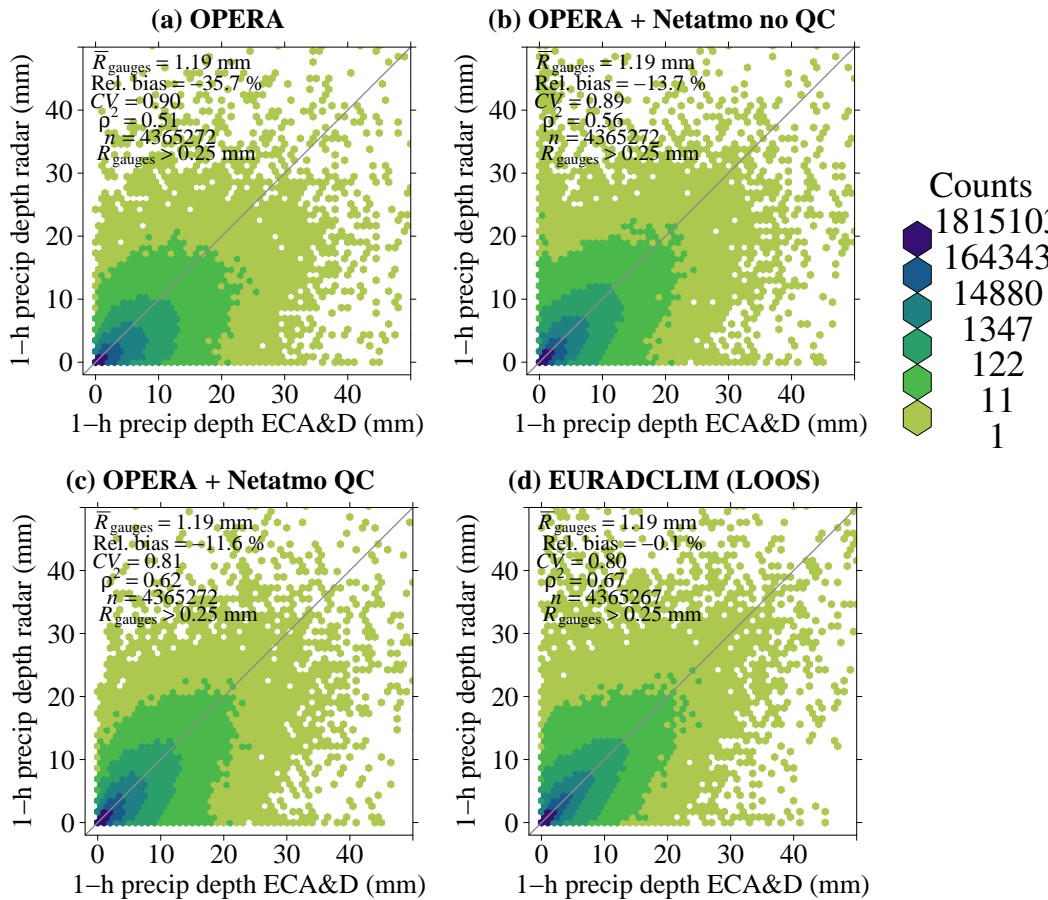

**Figure 4.** Scatter density plots of 1-h radar precipitation accumulations against disaggregated ECA&D rain gauge precipitation accumulations over the period September 1, 2019–August 31, 2020. Results are shown for the unadjusted dataset (a), for the merged OPERA-Netatmo datasets without (b) and with (c) quality control (QC) on Netatmo gauge data, and for the gauge-adjusted EURADCLIM dataset (d). The independent verification for EURADCLIM is performed via leave-one-out statistics (LOOS).

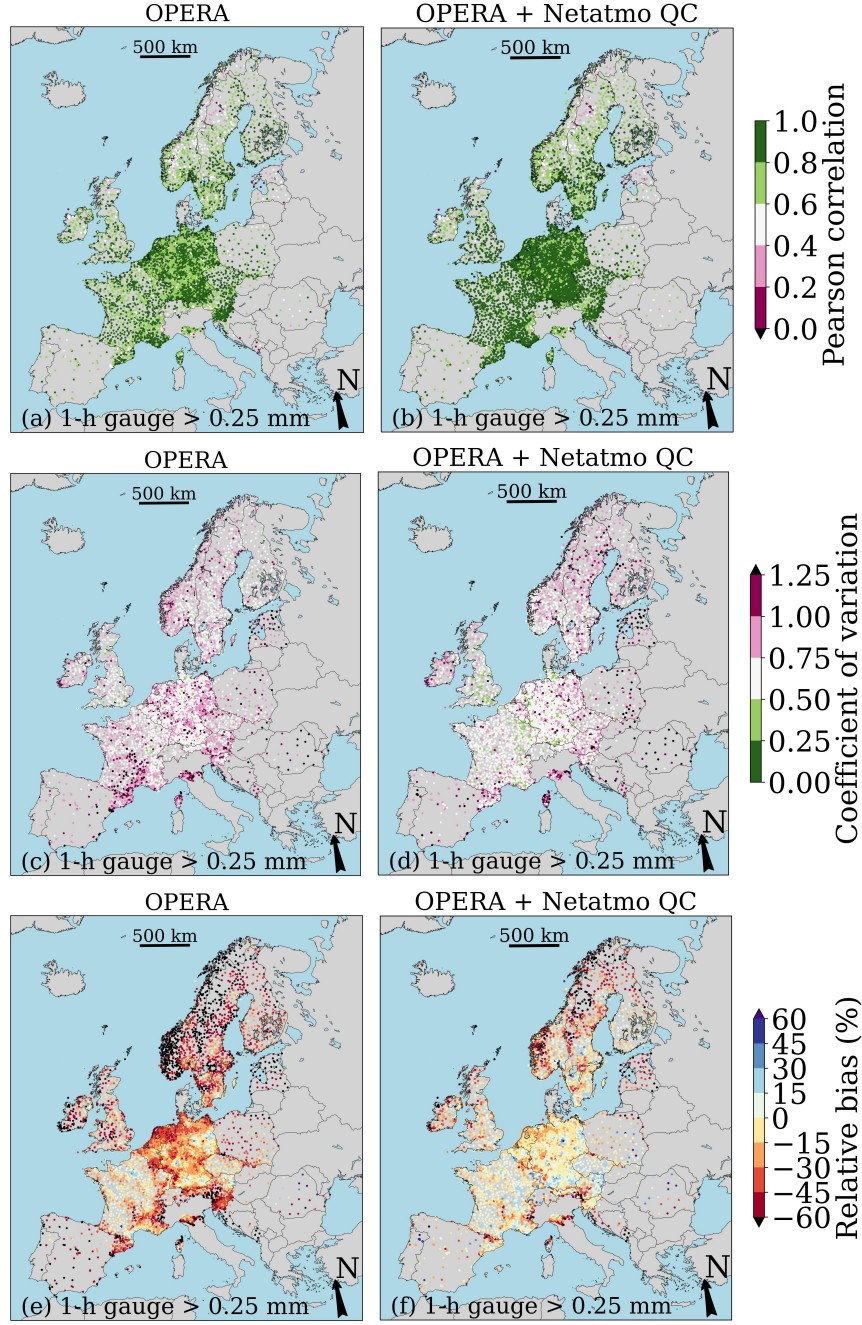

**Figure 5.** Spatial evaluation of 1-h precipitation accumulations against disaggregated clock-hourly ECA&D gauge precipitation accumulations over the period September 1, 2019–August 31, 2020. As indicated in Figure 1a, fewer ECA&D data are available for some gauge locations and hence the evaluation period is shorter. Verification results are displayed for the unadjusted OPERA radar dataset (a, c, e) and for the gauge-adjusted OPERA-Netatmo dataset (b, d, f). Maps made with Natural Earth. Free vector and raster map data ©naturalearthdata.com.





For each ECA&D rain gauge location a spatial verification is performed on 1-h accumulations for the unadjusted and Netatmo-adjusted (with QC) OPERA radar datasets (Figure 5). The quality of the unadjusted radar dataset is spatially variable, especially for CV and relative bias. The values for $\rho$ are reasonably high for the OPERA dataset, but clearly improve after merging. The improvement appears higher than found in Table 2 because of the discrete and relatively wide classes used in
Figure 5. Locally, clear improvements in the values for CV are found, although these can still be quite high. The relative bias strongly improves for the merged OPERA-Netatmo dataset compared to the unadjusted OPERA dataset and its spatial variability diminishes. Regionally, lower Netatmo gauge network densities may be a reason for poorer performance (e.g., Poland, Romania).

The spatial verification reveals severe underestimations for some regions, especially in Scandinavia. This may be related
to colder climates. The more frequent occurrence of solid precipitation may result in undercatch or delayed measurements by PWS gauges, because precipitation is not melted by a heating device. The manufacturer's specifications even report an operating temperature from 0°C to 50°C (Netatmo, 2022). To study a possible relationship between performance of merged radar precipitation accumulations and precipitation type, the evaluation is split into two: a group with lower (<5°C) and higher (≥5°C) daily average temperatures from the gridded E-OBS dataset. Because only daily averages of temperature are
available a relatively high threshold of 5°C has been used to account for the possibility of 1-h solid precipitation occurring at temperatures below this daily average. Scatter density plots presented in Figure 6 reveal a much more severe underestimation by the OPERA dataset at lower temperatures than at higher temperatures. This is expected given the generally higher influence of the vertical profile of reflectivity in radar QPE and the use of retrieval relations for rainfall in snow conditions. Much better results are obtained for higher temperatures. The same holds for the Netatmo-adjusted OPERA dataset. For lower temperatures,
the underestimation is less severe than the unadjusted OPERA dataset, but still a strong underestimation of 27% is found. This is 21 percentage points lower than the underestimation for higher temperatures, suggesting that the quality of Netatmo rainfall measurements declines for lower temperatures when the probability of solid precipitation is higher. The EURADCLIM dataset still reveals a better performance for higher temperatures, but the difference in relative bias is small between the two groups. Rain gauges are generally expected to have either a heating device (in case of automatic gauges) or instructions to melt
precipitation (in case of manual gauges).

Figure 7 shows annual precipitation sums for the OPERA dataset without and with Netatmo-adjustment. A comparison against the EURADCLIM dataset is also presented to assess the performance over the entire radar grid. The annual precipitation for the merged OPERA-Netatmo dataset is much larger than the unadjusted OPERA precipitation sum (Figure 7 a–b). The ratio between the merged OPERA-Netatmo dataset and the EURADCLIM dataset (Figure 7 d) is between 0.8–1.1, confirming its
quality. For some regions, a large underestimation with respect to EURADCLIM is found. This may partly be related to solid precipitation (Scandinavia), which is not expected to be captured well by Netatmo gauges. The limited spatial extent of the employed adjustment method may also result in an underestimation at locations far(ther) away from radars and rain gauges (e.g., above sea around the Iberian peninsula and Scotland).



**Figure 6.** Scatter density plots of 1-h radar precipitation accumulations against ECA&D rain gauges over the period September 1, 2019– August 31, 2020. Disaggregated clock-hourly gauge precipitation accumulations are employed. Results are shown for the unadjusted OPERA dataset (a–b), for a merged OPERA-Netatmo dataset (c–d), and for the EURADCLIM dataset (e–f). For each dataset, results are shown for 1-h accumulations on days with mean daily air temperature $<5°C$ (a, c, e) and $\geq 5°C$ (b, d, f). The independent verification for EURADCLIM is performed via leave-one-out statistics (LOOS).

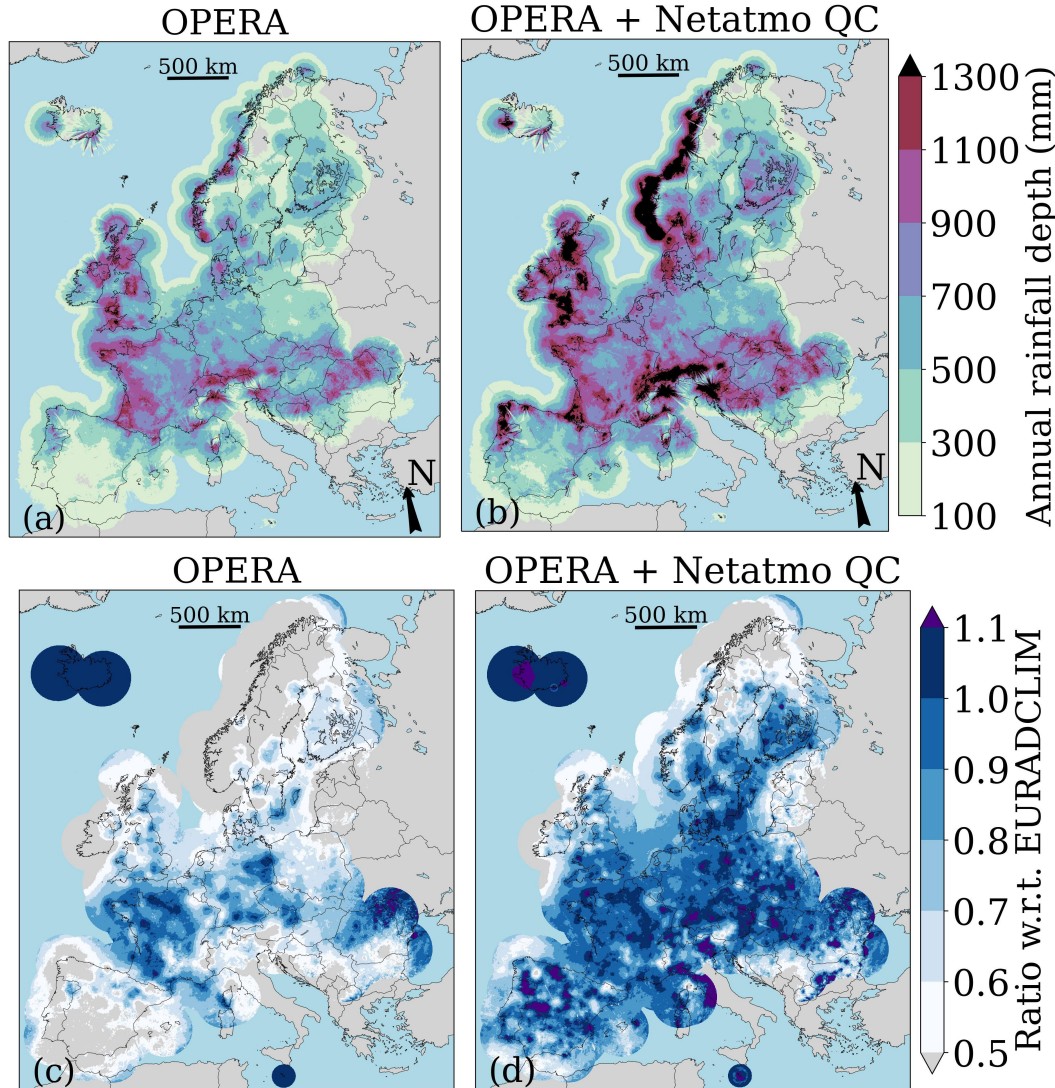

**Figure 7.** Maps with annual precipitation accumulations over the period September 1, 2019–August 31, 2020 for unadjusted OPERA (a) and the merged OPERA-Netatmo dataset (b). Maps with ratios of annual precipitation accumulations for both datasets with respect to EURADCLIM (c–d). Map made with Natural Earth. Free vector and raster map data ©naturalearthdata.com.

## 4.2 Extreme regional case study

Figure 8 illustrates the potential of merging with PWS gauge accumulations for extreme precipitation monitoring. It shows precipitation accumulations for a widespread event over northeastern Spain, with large areas with daily accumulations exceed-





ing 50 mm (panel c), and locally more than 150 mm (panel f). This very heavy rainfall event led to a strong rise in discharge of many small rivers in the south of Catalonia, causing four deaths and an estimated damage over €700,000 (Rigo et al., 2021). The large underestimation by OPERA (panel a) mainly disappears for the merged OPERA-Netatmo dataset (panel b) given its
resemblance with EURADCLIM (panel c). The underlying Netatmo gauge values (panel e) are in reasonable agreement with the ECA&D gauge values, although underestimations are found (panel f). Moreover, the density of the PWS network is higher than that of the ECA&D network (panel d) and the data could potentially be available in real-time, which is usually not the case for gauges in the ECA&D dataset. Note that for this region, the ECA&D network is rather dense compared to most of Europe and real-time data from these gauges are employed by the Meteorological Service of Catalonia for merging with radar data.
Note that only gauge locations (and values) are plotted if more than 0.25 mm of rain has been reported in (at least) one hour. Zooming in to a clock-hour in this 24-h period, a squall line becomes much more pronounced in the two merged radar datasets (panels h and i) compared to the unadjusted OPERA dataset (panel g). The Netatmo network provides more measurements (panel j), and this results in better sampling of the maximum precipitation within this hour (panels k and l). The squall line is more severe for EURADCLIM, though, possibly because of the local mean-field bias adjustment, which precedes the spatial
adjustment for EURADCLIM only. It is difficult to tell which dataset has the most realistic 1-h precipitation.

Rigo et al. (2021) provide gauge-adjusted precipitation estimates for the event presented in Fig. 8 employing data from four local C-band radars, which are not incorporated in the OPERA-based datasets. Hence, the distance to radars is much shorter than for the OPERA datasets, and precipitation is detected at a lower altitude. Rigo et al. (2021) present a real-time radar product which includes data from the gauges contributing to ECA&D, as well as a climatological product where the gauge data have
undergone additional QC. Both the merged OPERA-Netatmo dataset and EURADCLIM dataset underestimate precipitation compared to the local gauge-adjusted radar products. Hence, Fig. 8 does not provide the best possible precipitation estimates, but it does confirm that merging with PWS gauge data gives a large improvement in OPERA radar precipitation accumulations.

### 4.2.1 Extreme urban case study

The potential of PWS gauge accumulations for improving radar precipitation accumulations is expected to be higher in urban
areas. It can be more difficult to find an appropriate setup for rain gauges in urban environments. Hence, to comply with World Meteorological Organization regulations, NMHSs typically measure in areas with few or lower obstacles. Hence, the (automatic) weather stations are typically located outside city centres, often in more rural areas. In contrast, urban areas generally have the highest PWS network densities. Figure 9 illustrates an extreme 1-h precipitation event for the city of Brussels, Belgium. There are more than 80 Netatmo rain gauges in this area, whereas there are only 2 ECA&D gauges. The precipitation
accumulations increase for the EURADCLIM dataset (panel c) with respect to the unadjusted OPERA dataset (panel a). The increase is much stronger for the OPERA-Netatmo dataset (panel b), and it has the highest 1-h accumulation, 20 mm. The Netatmo rain gauges measure high accumulations (panel e). The ECA&D rain gauges only provide limited coverage and their values are lower than those from the closest PWS gauges (panels d–f). The high network density of PWS gauges helps to capture the localised extreme precipitation (panel e), which is hence reflected in the OPERA-Netatmo dataset (panel b). The


PWS gauges can not be considered to be the ground truth. But the fact that most gauges observe high(er) accumulations gives confidence.

**Figure 8.** An extreme precipitation event over northeastern Spain from October 22, 2019 0000 UTC – October 23, 2019 0000 UTC (24-h precipitation accumulation; a–f) and for 1800–1900 UTC within this period (g–l; 1-h accumulation). Accumulations are shown for the unadjusted OPERA (a, g), merged OPERA-Netatmo (b, h) and EURADCLIM (f, l) datasets. Locations of Netatmo and ECA&D gauges and their values are also provided (d–f, j–l). Map data ©OpenStreetMap contributors 2023. Distributed under the Open Data Commons Open Database License (ODbL) v1.0.

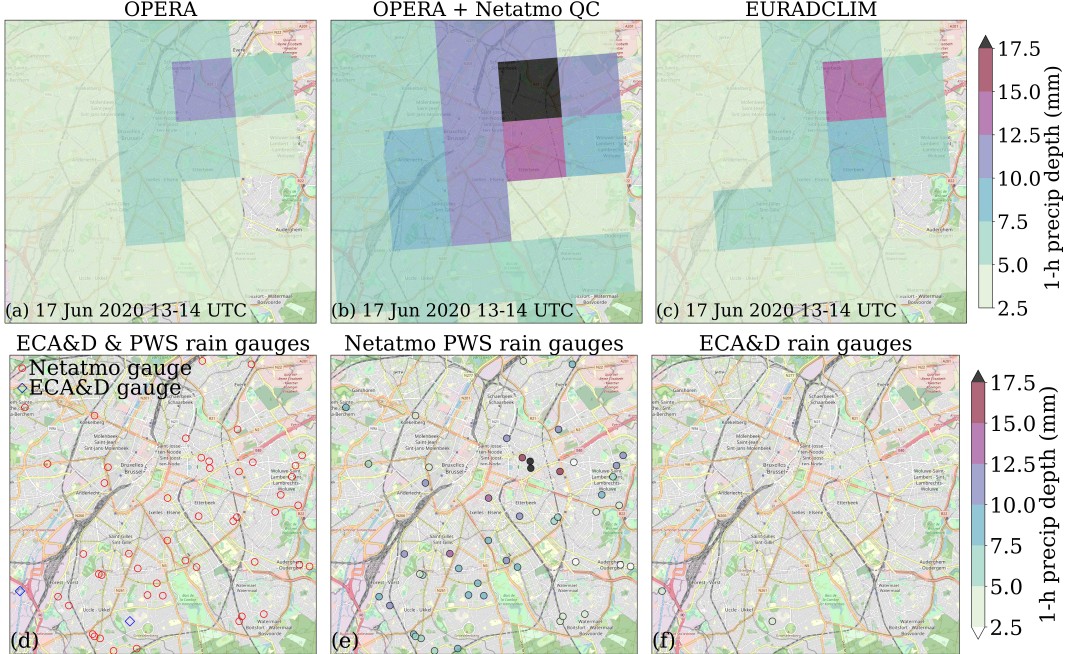

**Figure 9.** An extreme urban precipitation event over the city of Brussels, Belgium, from June 17, 2020 1300–1400 UTC (1-h precipitation accumulation). Accumulations are shown for the unadjusted OPERA (a), merged OPERA-Netatmo (b) and EURADCLIM (c) datasets. Locations of Netatmo and ECA&D gauges and their values are also provided (d–f). Map data ©OpenStreetMap contributors 2023. Distributed under the Open Data Commons Open Database License (ODbL) v1.0.

## 5 Discussion and recommendations

### 5.1 Suitability of reference data

The ECA&D rain gauge dataset is used as a reference dataset because of its established quality and its independence. The
disaggregated 1-h data are less independent because the OPERA radar data have been used for disaggregation. This may partly explain that for the OPERA-Netatmo dataset of 1-h accumulations, $\rho$ only improves moderately and the value of CV only improves for a gauge threshold of 0.25 mm. Since EURADCLIM already uses OPERA, there is also some dependence with the merged OPERA-Netatmo dataset. A disadvantage of using gauges as reference is the mismatch in sampling areas (a few dm$^2$ for gauges and 4-km$^2$ grid cells for OPERA) between the two datasets. This and other representativeness errors can be
large when radar and gauge accumulations are compared (Kitchen and Blackall, 1992), especially for sub-daily accumulations. Hence, differences between (merged) radar and gauge accumulations are not solely caused by sources of error in QPE.

The comparison with the performance of the radar dataset EURADCLIM is provided to put the quality of the merged radar-PWS dataset into perspective compared to a climatological dataset employing official rain gauges. The direct comparison with





EURADCLIM for annual precipitation and a case study allows for a high-resolution spatial comparison. The performance of
the merged OPERA-Netatmo dataset could also be compared to that of the near real-time OPERA-based radar rainfall product
for the European Rainfall-InduCed Hazard Assessment (ERICHA) system (Park et al., 2019). This would give more insight in
the potential of crowdsourcing for improving near real-time pan-European radar data. This is outside the scope of this paper,
but could be investigated in future research.

## 5.2 Improving quality control of PWS rain gauge data

This study should be seen as a starting point for merging PWS and radar precipitation accumulations. As shown in Fig. 4
and Tables 1 and 2, the QC of PWS rain gauge data has a positive effect, but with a relatively limited magnitude. A possible
explanation for this is that the merging algorithm already acts as a QC by reducing the impact of outliers. Multiple radar-gauge
pairs are employed to compute the adjustment factor for a given pixel, thus limiting the influence of a single gauge. Moreover,
only radar-gauge pairs are used for which radar and PWS both reported more than 0.25 mm in one hour, which helps to reduce
erroneous values. The quality of the merged product could potentially be further improved by applying additional QC to the
PWS gauge data. For instance, by applying a station outlier filter and a dynamically updated bias correction factor (De Vos
et al., 2019). Note that the radar-based part of the QC developed in this study (limits on the PWS/radar ratio and maximum
difference between PWS and radar accumulations), implicitly acts as a station outlier filter. Moreover, the performance of this
radar-based filter does not depend on the local PWS network density. The suggested station outlier filter (De Vos et al., 2019)
compares time series (by computing correlations) from neighbouring stations from at least the last two weeks (and longer if
there has been insufficient precipitation in this period) to detect deviations in local rainfall dynamics. These deviations could
be caused by, e.g., incorrect station locations.

Network density can affect the performance of a QC method, and whether a QC method is suitable at all. Van Andel (2021b)
developed a radar version of the QC (Van Andel, 2021a) developed by De Vos et al. (2019) and applied it to different rain gauge
datasets, including a 1 year Netatmo PWS gauge dataset over the Netherlands. This version of the QC algorithm performs well
in case of lower gauge density. Depending on the choosing setting (flex or strict filtering), results are not as good or similar for
an area with high network density (the Amsterdam metropolitan area) compared to applying the original PWS-based algorithm.
Moreover, this radar-based QC algorithm removes twice as much observations. But its potential for use in regions with lower
PWS network densities, such as commonly found in the European dataset, warrants further investigation.

Also note that the default parameter values from De Vos et al. (2019) for QC, based on Netatmo rain gauge data from the
Netherlands, are applied to data from the whole of Europe. Different parameter values could be tested on a European scale or
could be regionalised to take into account local precipitation climatology and local network characteristics. For instance, the
range with neighbouring PWSs is set to 10 km, but could be longer for stratiform rainfall. The same holds for regions with low
PWS network density.

Considering other QC algorithms, specifically the spatial consistency test against neighbours (SCT) could be relevant for
application to the PWS dataset used in this study, because it assesses the likelihood of an observation with stricter testing in case
of higher station density (Lussana et al., 2010; Alerskans et al., 2022) . The temporal consistency test and the spatial conformity





test from Ośródka et al. (2022) could be interesting to apply on PWS rain gauge data. Especially, the assignment of a quality index to each gauge observation is methodologically interesting. Only observations with a sufficiently high quality index could
be selected for merging with radar data, or the quality index could be used as weighting factor in the radar-PWS merging. The QC in this study mainly consists of inter-station checks and the use of an auxiliary source: radar data. The development and application of time series analyses from single stations (intra-station) checks may be explored to improve QC.

### 5.3 Bias correction of PWS and merged radar-PWS data

The quality of PWS gauge observations and sources of errors in radar QPE can be highly variable in space and time. This results
in spatial variability of the relative bias in 1-h merged precipitation accumulations (Figure 5). The quality of PWS gauge data could be improved by computation of a dynamically updated bias correction factor per gauge, employing neighbouring PWSs. This is intertwined with the station outlier filter (De Vos et al., 2019), as is the case for the radar version (Van Andel, 2021b). An extension to this could be to remove a gauge if the bias correction is smaller than 0.5 or greater than 2 (Van Andel, 2021b). The approach by Bárdossy et al. (2021), who apply bias correction via quantile mapping employing rain gauge observations
from the German Weather Service, is also worth investigating.

A seasonally and regionally variable bias correction factor could be applied to the merged radar-PWS dataset by using the methodology of Imhoff et al. (2021). This bias correction field can be derived from comparing with a reference (the EURADCLIM dataset) from previous years. This approach assumes that the quality of EURADCLIM is high across Europe and stationary in time. Improvements in OPERA precipitation data would require recalculation of bias correction factors.
Applying this approach seems challenging, given the large improvement found for daily unadjusted OPERA accumulations for the warm seasons from 2015 to 2018 (Park et al., 2019).

### 5.4 Adaptive merging of radar and PWS data

Another possible improvement in the merged radar-PWS dataset could be obtained by the application or development of other adjustment methods (see e.g. Goudenhoofdt and Delobbe, 2009). Or, for instance, by taking into account the local gauge
network density and local precipitation climatology (instead of the climatology from the Netherlands) to select radar-gauge pairs employed to adjust radar precipitation for a given grid cell. Improving the climatological gauge-adjusted radar dataset EURADCLIM by merging both the PWS and the ECA&D rain gauge data at once is another avenue that can be explored. The merging algorithm employed in this study can use quality information about the gauge observations, giving lower weight to lower-quality gauges (in this case the PWS gauges).

### 5.5 Opportunities for merged precipitation products

In addition to Netatmo PWS gauge data, data from other PWS platforms and companies could also potentially be taken into account for merging with radar data. Examples of such platforms are the Weather Observations Website (WOW; free access; https://wow.metoffice.gov.uk/; https://wow.knmi.nl/; O'Hara et al. (2023)), the Weather Underground website (https://www.





wunderground.com/wundermap) (Bell et al., 2013; Kirk et al., 2021), Weathercloud (https://app.weathercloud.net/map#), Me-
teonetwork (https://www.meteonetwork.it/rete/livemap/ and https://meteonetwork.eu/en), PWSweather (https://www.pwsweather.
com/map) and (solely for daily precipitation in the United States of America) the Community Collaborative Rain, Hail
and Snow Network (CoCoRaHS; https://www.cocorahs.org/) (Reges et al., 2016). The sensor quality and setup of PWSs
for those data sources can be better compared to Netatmo PWSs. For WOW, station owners can fill in location attributes
related to the surroundings, measurement device and timeliness of reporting. These are employed to compute site ratings
(https://wow.metoffice.gov.uk/support/siteratings), which could be used as part of the QC on PWS data or as quality index in
radar-PWS merging. For Meteonetwork, setup guidelines need to be followed for a station to be accepted. Automatic QC is
applied to data from Italy (Giazzi et al., 2022).

   PWS air temperature measurements (Bárdossy et al., 2021) or air temperatures forecasts (Båserud et al., 2020) could be used
to estimate whether precipitation is solid or liquid, and could hence be used to select which gauges to use in merging. This
could prevent the influence of undercatch (particularly in high-wind situations) of snow or delayed measurements due to the
fact that frozen precipitation is only registered when it melts (see also Fig. 6).

   Real-time data from official networks of automatic rain gauges are generally much sparser and a latency of 5–30 min is
not uncommon. Several crowdsourced rain gauge networks potentially disseminate data with similar or even shorter latency.
Arrangement of data contracts with commercial companies, preferably on a European or even global level, could help making
these data available for everyone (either directly or indirectly through merging in precipitation products). In return, these
companies would receive valuable feedback on the quality of their data. And the availability of these data could spur further
development of QC algorithms by the scientific community that could then be employed by these companies for their platforms.

   Commercial microwave link (CML) data (Messer et al., 2006; Leijnse et al., 2007; Overeem et al., 2016; Graf et al., 2020)
could also be considered as a candidate for merging with radar data. Quality control, rainfall retrieval, and merging of oppor-
tunistic sensing data, including PWSs, CMLs and SMLs (satellite microwave links), are being studied in the OpenSense (Op-
portunistic Precipitation Sensing Network; https://opensenseaction.eu/) COST Action. OpenSense will address benchmarking
of QC algorithms on datasets, allowing for a fair and standardised comparison of QC algorithms. Development of operational
merged products based on radar, PWS or CML data is already being investigated by some NMHSs in Europe (Wenzel et al.,
2023). For a specific country or region, it may be efficient to focus on the most promising opportunistic data sources: those for
which network densities are highest, obtained experiences and results are favourable, and the probability of continuous data
access is higher.

   Merging opportunistic sensing data with radar data also involves risks and creates dependence on data beyond control of
NMHSs. For instance, PWS sensors and setups could degrade over the course of time and network densities may decrease. As
a result, the quality of merged products can decrease. Methods that are employed to merge such data with radar accumulations
should therefore be designed such that they can deal with varying quality and availability of opportunistic sensors. Another
way to try to prevent the decline in quality and number of opportunistic sensors is social engagement of citizen scientists and
outreach to increase the number of citizen scientists. Garcia-Marti et al. (2023) provide an overview of ways to achieve this.
Another issue that may play a role with the increased use of opportunistic sensors is the decrease in perceived importance of



data from official rain gauge networks, potentially leading to abandoning such networks. However, such high-quality station
data are indispensable for climate monitoring. This should be kept in mind by agencies operating these networks.

## 6   Conclusions

A 1-year, quality-controlled, Netatmo PWS rain gauge dataset of 1-h accumulations was merged with 1-h OPERA radar accumulations over Europe. The PWS rain gauge data were subjected to QC employing neighbouring PWSs and unadjusted radar accumulations. A default bias correction factor of 1.063 was applied to the PWS accumulations. The potential of crowdsourced
data for improving radar precipitation products is confirmed by an evaluation against independent official rain gauges from ECA&D for hourly and daily precipitation. Underestimation for daily precipitation declines from ∼28% for the unadjusted radar dataset to ∼3% for the merged radar-PWS dataset (using all values). In some regions much stronger underestimations are found, which may be related to solid precipitation and lower PWS network densities. Underestimation of 1-h precipitation is ∼27% for mean daily air temperatures <5°C and ∼7% for mean daily air temperatures ≥5°C (only using values for which
the gauge value exceeds 0.25 mm). Underestimation is much less pronounced for the EURADCLIM dataset, which has been adjusted using data from official networks, and can largely correct for radar underestimation of solid precipitation. This suggests that solid precipitation is not properly captured by Netatmo PWSs, which limits the performance of a merged radar-PWS dataset.

The quality of the merged radar-PWS dataset is generally lower than that of the climatological EURADCLIM dataset.
When compared to daily ECA&D gauge accumulations, scatter is higher and an underestimation of ∼10% instead of an overestimation of 1% is found. Since the availability of (near) real-time rain gauge accumulations from official networks is usually much lower compared to climatological data, a (near) real-time merged OPERA-gauge product is not expected to achieve the same quality as EURADCLIM. The outcome of this study paves the way for (near) real-time merging of PWS and OPERA radar precipitation accumulations for the European continent, but its findings may also be applicable to other regions
with radar coverage, such as North America, or for national radar composites. The potential of such a merged dataset for nowcasting of precipitation could also be investigated at the continental scale. To conclude, PWS rain gauge data can add value to already high-resolution radar accumulations, by improving the quality of localised precipitation estimates. This potentially allows for better warnings of severe weather and associated (flash) floods and better-informed decision making for disaster risk management (Figure 9).

*Code and data availability.* The EURADCLIM 1-h and 24-h precipitation open datasets can be obtained from https://doi.org/10.21944/7ypj-wn68 (Overeem et al., 2022a) and https://doi.org/10.21944/1a54-gg96 (Overeem et al., 2022b). Unadjusted OPERA radar data are available for the research community after registration (https://www.eumetnet.eu/activities/observations-programme/current-activities/opera/). The gridded dataset of daily mean air temperature, E-OBS version 26.0e (release October 2022), is publicly available (Cornes et al., 2018; Copernicus Climate Change Service, 2022). Most daily rain gauge time series from the ECA&D project are publicly available
(https://www.ecad.eu). The Netatmo rain gauge dataset was purchased from commercial company Netatmo and can not be made publicly



available due to legal restrictions. The preprocessing and part of the quality control of the Netatmo gauge data, i.e., the faulty zeroes and high influx filters, are based on openly available code written in language R (https://github.com/LottedeVos/PWSQC). This was developed by De Vos et al. (2019), with some modifications to work with the data on a European level, and not including any computations with a reference dataset. Precipitation maps and maps showing the shortest distance to a gauge were made with publicly available code written in Python
(Overeem, 2022, 2023).

*Author contributions.* **Aart Overeem** conceptualisation, data curation, formal analysis, funding acquisition, investigation, methodology, project administration, software, supervision, validation, visualisation, writing – original draft preparation, writing – review & editing. **Hidde Leijnse** conceptualisation, funding acquisition, methodology, software, supervision, writing – original draft preparation, writing – review & editing. **Gerard van der Schrier** data curation, funding acquisition, investigation, writing – review & editing. **Else van den Besselaar**
data curation, investigation, writing – review & editing. **Irene Garcia-Marti** writing – review & editing. **Lotte de Vos** conceptualisation, methodology, software, writing – review & editing.

*Competing interests.* The authors declare that they have no conflict of interest.

*Acknowledgements.* We thank Met Norway for jointly purchasing the Netatmo rain gauge dataset. We acknowledge the private company Netatmo for providing the rain gauge data and for their cooperative attitude. This research would not have been possible without the com-
mitment of the citizen scientists providing data from their Netatmo IoT rain gauge devices, which they bought, installed, and maintained. We acknowledge the E-OBS dataset (Cornes et al., 2018) from the EU-FP6 project Uncertainties in Ensembles of Regional ReAnalyses (https://www.uerra.eu) and the Copernicus Climate Change Service, and the data providers in the ECA&D project (https://www.ecad.eu), as well as the NMHSs who sent radar data to OPERA. We thank Dr. Marc Prohom and Dr. Tomeu Rigo from the Meteorological Service of Catalonia, for providing additional information on the extreme precipitation event in Figure 8, and for reviewing our interpretation. This study
was funded by the Royal Netherlands Meteorological Institute in the framework of the EURADCLIM project (https://www.knmi.nl/research/observations-data-technology/projects/euradclim-the-european-climatological-high-resolution-gauge-adjusted-radar-rainfall-dataset). Maps were produced with Python package Cartopy (Met Office, 2022). Project EURADCLIM, which includes the work for this paper, was financed by KNMI's multi-annual strategic research programme (project number 2017.02).





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
