# Peer review of "Merging with crowdsourced rain gauge data improves pan-European radar precipitation estimates"

_Hydrology and Earth System Sciences, 2023_

## Referee Comment (RC1)

**Review of hess-2023-122: "Merging with crowdsourced rain gauge data improves pan-European radar precipitation estimates"**

**1 General comments**

The article evaluates how a pan-European quantitative precipitation estimation (QPE) based on OPERA radar rain rate adjusted with personal weather station (PWS) rain gauge observations compares with (1) a QPE based on OPERA radar rain rate adjusted with ECA&D rain gauge observations called EURADCLIM, (2) a QPE based on OPERA radar rain rate unadjusted with observations, when evaluated against ECA&D rain gauge observations.

The article is well-written and of good quality, I recommend this manuscript to be published after technical corrections and, if the authors find them useful, taking into account my two specific comments.

**2 Specific comments**

- L150 and L300: "the NMHSs that generally operate their rain gauges in open rural areas (following World Meteorological Organization regulations)." and "Hence, to comply with World Meteorological Organization regulations, NMHSs typically measure in areas with few or lower obstacles. Hence, the (automatic) weather stations are typically located outside city centres, often in more rural areas.". These two sentences suggest that the WMO's installation rules force NMHSs to install their stations outside urban centres. I'd like to qualify this statement: first, even if it is more complicated than in the countryside, it is possible to find open areas (for example in parks, sports complexes, etc.) in the urban centres of many towns and cities. Secondly, the installation of NMHS stations is historically linked, in the 20th century, to the needs of the aeronautical industry, and so near the big cities, the stations are often installed on airports. Finally, the densification of many European networks was often carried out to ensure regular coverage of countries, and the city centres represent negligible fractions of the area of many countries.

- L265: "Rain gauges are generally expected to have either a heating device (in case of automatic gauges) or instructions to melt precipitation (in case of manual gauges)." Also, NMHS rain gauges often have a larger reception cone volume than Netatmo rain gauges, enabling them to collect more solid precipitation, which gradually melt after falling, even if they are not heated.

**3 Technical corrections**

- L5 and following: "1-year" should be "1 year". Consider following the HESS submission guidelines (https://www.hydrology-and-earth-system-sciences.net/submission.html) for hyphens.

- L68 and following: "September 1, 2019" should be "1 September 2019". Consider following the HESS guidelines for date and time.

- L117: "and coauthors" -> "et al."?

- L146 and following: "Figure 1c–d" -> "Fig. 1c–d". According to the HESS guidelines: "The abbreviation "Fig." should be used when it appears in running text and should be followed by a number unless it comes at the beginning of a sentence."

- L274 and following: I find the word "extreme" a bit too much for the examples given. "Heavy rainfall" is sufficient. The threshold of 150 mm in 1 day is the usual threshold in the Mediterranean basin for what is known in the literature as a heavy precipitation event (HPE).

- L277 and following: (panel c) -> (Fig. 8c)?

- L484 and L526: consider providing Digital Object Identifier (DOI) for these two articles that have one.

- L535: "https://doi.org/https://doi.org/" should be "https://doi.org/".

- L543: "https://www.netatmo.com/en-gb/weather/weatherstation/specifications" should be updated to "https://www.netatmo.com/en-eu/smart-weather-station#specifications".

- L569: "https://knmi-ecad-assets-prd.s3.amazonaws.com/documents/atbd.pdf" is not working.

---

## Referee Comment (RC2)

**Review: "Merging with crowdsourced rain gauge data improves pan-European radar precipitation estimates",**

**by Aart Overeem et al.**

My overall evaluation of the article presented to me for review is high. The few comments I have written below are rather secondary and are more questions to the Authors, who may not necessarily agree with them.

The main strength of the paper in my opinion is that it addresses the latest developments in QPE and QC of rainfall data on European ground.

**Most important doubts (questions):**

- Why were only OPERA and Netatmo data used, without rain gauges of even a few NMHSs? More rain gauges = their QC is more effective.

- Does the scheme shown include the use of some form of quality index (QI) or quality flag? E.g. in the case of 'flex filtering', when there are less than 5 other rain gauges in a neighbourhood, then one could keep the value in a given rain gauge but lower its QI. If we included the QI in these algorithms, then that rainfall height would enter into spatial interpolation and/or merging, but with lower weight.

- P. 9, l. 178-180: What about the reverse case, where the radar does not see the weak precipitation found by the rain gauge(s), which happens at greater distances from the radar site as a result of a radar beam overshooting the precipitation? This happens when the rainfall is from low clouds, especially in colder periods.

- P. 9-10, Sect. 3: How about providing the full formulas for the statistical metrics used? This always makes analysis easier.

- Table 1: Are these numbers in the top row of the table for 'no threshold'? It might be worth writing this in the relevant lines of the table.

- Table 1 and p. 11, 222-223: How to interpret this table? Do the small differences between the PWS data without and with QC mean that the PWS data are good on their own, or rather that

QC is too ineffective (or too moderate)? The former possibility might be suggested by the large improvement in the "OPERA + Netatmo No QC" data relative to "OPERA".

- P. 15, l. 249-265: For me, these are very valuable insights!

- Fig. 8: What are the benefits of using the PWS data in this figure? The spatial distribution of the precipitation field is very similar in the three maps shown, so is the scaling the main benefit? This is what the commentary in l. 275-289 suggests as well, so why not also present such statistics to show this impact on the distribution? E.g., the correlation coefficient...

- Fig. 9 is also very extremely interesting!

Good luck!

---

## Author Comment (AC1)

Reply to comments by Anonymous Referee #1.

**Our response to the comments by Referee #1 is provided in bold font.**

*1. General Comments*

The article evaluates how a pan-European quantitative precipitation estimation (QPE) based on OPERA radar rain rate adjusted with personal weather station (PWS) rain gauge observations compares with (1) a QPE based on OPERA radar rain rate adjusted with ECA&D rain gauge observations called EURADCLIM, (2) a QPE based on OPERA radar rain rate unadjusted with observations, when evaluated against ECA&D rain gauge observations.

The article is well-written and of good quality, I recommend this manuscript to be published after technical corrections and, if the authors find them useful, taking into account my two specific comments.

**We thank the reviewer for recognizing the value of our work and for the constructive feedback. The suggestions to modify the paper in accordance with the HESS submission guidelines were appreciated.**

*2. Specific Comments*

– L150 and L300: "the NMHSs that generally operate their rain gauges in open rural areas (following World Meteorological Organization regulations)." and "Hence, to comply with World Meteorological Organization regulations, NMHSs typically measure in areas with few or lower obstacles. Hence, the (automatic) weather stations are typically located outside city centres, often in more rural areas.". These two sentences suggest that the WMO's installation rules force NMHSs to install their stations outside urban centres. I'd like to qualify this statement: first, even if it is more complicated than in the countryside, it is possible to find open areas (for example in parks, sports complexes, etc.) in the urban centres of many towns and cities. Secondly, the installation of NMHS stations is historically linked, in the 20th century, to the needs of the aeronautical industry, and so near the big cities, the stations are often installed on airports. Finally, the densification of many European networks was often carried out to ensure regular coverage of countries, and the city centres represent negligible fractions of the area of many countries.

**We modified these statements and replaced 1) "Note that the PWS network tends to be of high density in the areas where the population density is high, which contrasts with the networks operated by the NMHSs that generally operate their rain gauges in open rural areas (following World Meteorological Organization regulations)." by (L. 149-151) "Note that the PWS network tends to be of high density in the areas where the population density is high, whereas it is more**

**difficult to find locations for NMHS gauges in urban areas complying with World Meteorological Organization regulations.”; 2) “Hence, to comply with World Meteorological Organization regulations, NMHSs typically measure in areas with few or lower obstacles. Hence, the (automatic) weather stations are typically located outside city centres, often in more rural areas.” by (L. 315-320) “Hence, to comply with World Meteorological Organization regulations, NMHSs typically measure in areas with few or lower obstacles. Therefore, the (automatic) weather stations are typically located outside city centres, often in more rural areas, although open areas can also be found in cities. Note that originally NMHS weather stations were often installed at airports, close to large cities, and are present to date. Although urban rainfall monitoring is relevant, cities generally represent only a small fraction of the land surface of a country. Meteorological networks have been designed to ensure regular coverage of countries.”**

- L265: "Rain gauges are generally expected to have either a heating device (in case of automatic gauges) or instructions to melt precipitation (in case of manual gauges)." Also, NMHS rain gauges often have a larger reception cone volume than Netatmo rain gauges, enabling them to collect more solid precipitation, which gradually melt after falling, even if they are not heated.

**We added to the text (L. 278-280): “Moreover, rain gauges operated by NMHSs often have a larger reception cone volume than PWS rain gauges, enabling them to collect more solid precipitation, which can gradually melt, even if the gauges are not heated.”**

*3. Technical Corrections*

- L5 and following: "1-year" should be "1 year". Consider following the HESS submission guidelines (https://www.hydrology-and-earth-system-sciences.net/submission.html) for hyphens.

**We removed all these kind of occurrences of hyphens, e.g., for “1-year”, “1-h”, “24-h”, et cetera, also in the figures.**

- L68 and following: "September 1, 2019" should be "1 September 2019". Consider following the HESS guidelines for date and time.

**We changed this accordingly for all occurrences of dates.**

- L117: "and coauthors" -> "et al."?

**We now list all 39 coauthors in the reference list.**

- L146 and following: "Figure 1c–d" -> "Fig. 1c–d". According to the HESS guidelines: "The abbreviation "Fig." should be used when it appears in running text and should be followed by a number unless it comes at the beginning of a sentence."

**We modified this accordingly.**

- L274 and following: I find the word "extreme" a bit too much for the examples given. "Heavy rainfall" is sufficient. The threshold of 150 mm in 1 day is the usual threshold in the Mediterranean basin for what is known in the literature as a heavy precipitation event (HPE).

**We replaced "extreme" by "heavy rainfall" or "heavy precipitation".**

- L277 and following: (panel c) -> (Fig. 8c)?

**We replaced all occurrences of "panel" by "Fig. 8" or "Fig. 9".**

- L484 and L526: consider providing Digital Object Identifier (DOI) for these two articles that have one.

**We added a DOI for these two articles.**

- L535: "https://doi.org/https://doi.org/" should be "https://doi.org/".

**We corrected this.**

- L543: "https://www.netatmo.com/en-gb/weather/weatherstation/specifications" should be updated to "https://www.netatmo.com/en-eu/smart-weather-station#specifications".

**Because we specifically refer to the specifications of the rain gauge module, we now refer to "https://www.netatmo.com/en-eu/smart-rain-gauge".**

- L569: "https://knmi-ecad-assets-prd.s3.amazonaws.com/documents/atbd.pdf" is not working.

**The correct URL was in the bibliography file, but when clicking on the link in the PDF document it did not work indeed. We solved this by putting the URL on the next line.**

---

## Author Comment (AC2)

Reply to comments by Anonymous Referee #2.

**Our response to the comments by Referee #2 is provided in bold font.**

My overall evaluation of the article presented to me for review is high. The few comments I have written below are rather secondary and are more questions to the Authors, who may not necessarily agree with them.

The main strength of the paper in my opinion is that it addresses the latest developments in QPE and QC of rainfall data on European ground.

**We thank the reviewer for providing a positive and constructive assessment of our manuscript.**

*Most important doubts (questions):*

- Why were only OPERA and Netatmo data used, without rain gauges of even a few NMHSs? More rain gauges = their QC is more effective.

**The reviewer is right that ideally all available rain gauge data would be combined. This could be beneficial for the QC of PWS rain gauge data, but also to obtain the best possible gauge-adjusted radar precipitation accumulations (i.e., for the merging itself). Concerning the latter, we already mention "Improving the climatological gauge-adjusted radar dataset EURADCLIM by merging both the PWS and the ECA&D rain gauge data at once is another avenue that can be explored.". Since the aim of this study is to show the potential of PWS gauge observations for improving radar data, we decided to use the NMHS gauges only for an independent evaluation. We added to the manuscript (L. 402-403): "Finally, NMHS rain gauge data could also be taken into account in the quality control of PWS rain gauge data.".**

- Does the scheme shown include the use of some form of quality index (QI) or quality flag? E.g. in the case of 'flex filtering', when there are less than 5 other rain gauges in a neighbourhood, then one could keep the value in a given rain gauge but lower its QI. If we included the QI in these algorithms, then that rainfall height would enter into spatial interpolation and/or merging, but with lower weight.

**Currently, the scheme assumes all gauge data to have the same quality, but, as already mentioned in the manuscript, "The merging algorithm employed in this study can use quality information about the gauge observations, giving lower weight to lower-quality gauges (in this case the PWS gauges).". We agree that incorporating quality information on PWS rain gauge data would be relevant, where its weighing would, for instance, depend on the outcome of the quality control and the type of rain gauge being used. This will require a thorough evaluation of the appropriate weights that should be used given the outcome of the quality control. Especially, when NMHS and PWS rain gauge data would be combined in the merging, assignment of**

**weighing factors will become even more relevant to avoid that the more accurate NMHS gauge data would be overwhelmed by the many more observations from the less accurate PWS gauges.**

- P. 9, l. 178-180: What about the reverse case, where the radar does not see the weak precipitation found by the rain gauge(s), which happens at greater distances from the radar site as a result of a radar beam overshooting the precipitation? This happens when the rainfall is from low clouds, especially in colder periods.

**It could indeed occur that the radar accumulation becomes lower than 0.25 mm in 1 hour due to overshooting of precipitation, implying that the radar-gauge pair is not used to compute an adjustment factor field. Note that in case precipitation is entirely missed in the radar image, the radar accumulation of zero cannot be increased anyway.**

- P. 9-10, Sect. 3: How about providing the full formulas for the statistical metrics used? This always makes analysis easier.

**We added the equations for relative bias, coefficient of variation, Pearson correlation coefficient and mean absolute error to Section 3.**

- Table 1: Are these numbers in the top row of the table for 'no threshold'? It might be worth writing this in the relevant lines of the table.

**Yes, the numbers in the top row are for "no threshold". We added "No threshold" to the lines where no threshold is applied.**

- Table 1 and p. 11, 222-223: How to interpret this table? Do the small differences between the PWS data without and with QC mean that the PWS data are good on their own, or rather that QC is too ineffective (or too moderate)? The former possibility might be suggested by the large improvement in the "OPERA + Netatmo No QC" data relative to "OPERA".

**It is probably a combination of the quality of the PWS data already being relatively good, and the merging algorithm acting as a kind of quality control. We added to the beginning of the Results Section (L. 228-230): "No QC implies that the quality control in Section 3.1 has been omitted, except that the 1 h PWS accumulations are used for merging if both the unadjusted radar value and the Netatmo gauge value are larger than 0.25 mm." (because it was not entirely clear from the manuscript what is meant with "No QC") & (L. 233-236) "The relatively good performance for No QC could be attributed to the quality of PWS gauge data and to the merging algorithm acting as kind of quality control. Only radar-gauge pairs are used in the merging for which radar and PWS observe more than 0.25 mm. Moreover, a spatial adjustment factor is computed by distance-weighted averaging of radar and PWS values, which can average out outliers.". Note that when we discuss the scatter density plots we do see a positive and**

**important impact of the quality control: "The scatter density plots also reveal how quality control has a positive impact: a group of large precipitation accumulations in case of lower gauge accumulations is removed.".**

- P. 15, l. 249-265: For me, these are very valuable insights!

**Thank you.**

- Fig. 8: What are the benefits of using the PWS data in this figure? The spatial distribution of the precipitation field is very similar in the three maps shown, so is the scaling the main benefit? This is what the commentary in l. 275-289 suggests as well, so why not also present such statistics to show this impact on the distribution? E.g., the correlation coefficient...

**We indeed do not show clear benefits of the quality of the PWS-based dataset. Given these results, presenting metrics will likely not provide new insights. There is, however, one clear benefit of the PWS-based dataset, that is already mentioned: "Moreover, the density of the PWS network is higher than that of the ECA\&D network (Fig. 8d) and the data could potentially be available in real-time, which is usually not the case for gauges in the ECA\&D dataset.".**

- Fig. 9 is also very extremely interesting!

**Thank you.**

Good luck!

---

## Author Response (AR2)

We thank the reviewers for their assessment and the editor for accepting our manuscript. Since we were allowed to publish the merged OPERA-Netatmo precipitation dataset, we added a reference to a repository in the code and data availability section, this being the only change with respect to the accepted version of our manuscript.